# A large-scale evaluation of NLP-derived chemical-gene/protein relationships from the scientific literature: Implications for knowledge graph construction

Jonathan C. G. Jeynes ⬢*, Matthew Corney, Tim James

Evotec (UK) Ltd., *in silico* Research and Development, Milton Park, Abingdon, Oxfordshire, United Kingdom

* charlie.jeynes@evotec.com

## Abstract

One area of active research is the use of natural language processing (NLP) to mine bio-medical texts for sets of triples (subject-predicate-object) for knowledge graph (KG) construction. While statistical methods to mine co-occurrences of entities within sentences are relatively robust, accurate relationship extraction is more challenging. Herein, we evaluate the Global Network of Biomedical Relationships (GNBR), a dataset that uses distributional semantics to model relationships between biomedical entities. The focus of our paper is an evaluation of a subset of the GNBR data; the relationships between chemicals and genes/proteins. We use Evotec's structured 'Nexus' database of >2.76M chemical-protein interactions as a ground truth to compare with GNBRs relationships and find a micro-averaged precision-recall area under the curve (AUC) of 0.50 and a micro-averaged receiver operating characteristic (ROC) curve AUC of 0.71 across the relationship classes 'inhibits', 'binding', 'agonism' and 'antagonism', when a comparison is made on a sentence-by-sentence basis. We conclude that, even though these micro-average scores are modest, using a high threshold on certain relationship classes like 'inhibits' could yield high fidelity triples that are not reported in structured datasets. We discuss how different methods of processing GNBR data, and the factuality of triples could affect the accuracy of NLP data incorporated into knowledge graphs. We provide a GNBR-Nexus(ChEMBL-subset) merged datafile that contains over 20,000 sentences where a protein/gene-chemical co-occur and includes both the GNBR relationship scores as well as the ChEMBL (manually curated) relationships (e.g., 'agonist', 'inhibitor') —this can be accessed at https://doi.org/10.5281/zenodo.8136752. We envisage this being used to aid curation efforts by the drug discovery community.

## Introduction

A recent trend within drug discovery is the construction of knowledge graphs combining disparate data sources to facilitate cross-domain analysis [1]. Beyond simply visualizing links between chemicals, genes and diseases, knowledge graphs can have considerable predictive

**Data Availability Statement:** The sources of data that were used in this study and the data repository we are releasing are here: 1. The "Global Network of Biomedical Relationships" repository available

here: https://doi.org/10.5281/zenodo.3459420 2.
Evotec's "Nexus" database cannot be shared
publicly because it contains data from third party
sources (e.g. Clarivate/Cortellis) where the licence
prohibits sharing. For more information about this
database please contact daniel.grindrod@evotec.
com. 3. We have also made a repository available
from the results of our study which is available
here: https://doi.org/10.5281/zenodo.8136752.

**Funding:** The author(s) received no specific
funding for this work.

**Competing interests:** I have read the journal's
policy and the authors of this manuscript have the
following competing interests: All authors are
employees of Evotec (UK) Ltd. This does not alter
our adherence to PLOS ONE policies on sharing
data and materials.

power once machine learning techniques have been applied to the data [2, 3]. Typically, this
data is formatted into sets of triples; a head, a relationship, and a tail entity, such as 'drug treats
disease' or 'chemical inhibits protein'. Within the biomedical domain there are a number of
high-quality, structured data sources, including UniProt, Gene Ontology, Open Targets, Drug-
Bank and ChEMBL [4]. Examples of published knowledge graphs include Cornell University's
biomedical knowledge graph, which contains information from 18 structured databases [5],
CROssBAR, which contains data from 14 [6] and SPOKE (previously called Hetionet [7]).
However, due to the mostly manual annotation process that many of these datasets rely on, rel-
evant information from the scientific literature is almost certainly missed.

To address the issue of missing data, a substantial effort has been made to automatically
capture biomedical information from the literature using natural language processing (NLP)
or text mining algorithms. Text mining is primarily concerned with detecting statistical co-
occurrences of biomedical entities within sentences using named entity recognition models.
Notably, PubTator [8] and Open Targets [9] both have freely accessible and frequently updated
co-occurrence datasets, with named entities mined from the entirety of PubMed full text arti-
cles and MEDLINE abstracts. There are also multiple commercial providers of NLP-based
relationship extraction data and software within biomedicine, including Causaly, Biorelate,
Linguamatics and SciBite. There are also other companies, such as BenevolentAI [10] and
Tellic [11], which amalgamate multiple curated data sources with their own NLP-derived data-
sets. Similarly, many large pharma companies have created their own knowledge graphs. For
instance, Astra Zeneca's 'Biomedical Insights Knowledge Graph (BIKG)' incorporates more
than 39 data sources, has around 11 million entities (nodes) and 118 million relationships
(edges), with a substantial proportion (80%) of those derived from their own NLP workflows
[12]. Another example is AbbVie/Tellic's knowledge graph that contains 911 million relation-
ships (edges) constructed from structured and NLP-derived data [11].

However, to our knowledge, there are only two publicly available datasets that contain
NLP-derived *relationships* that cover a broad range of biomedical entities (e.g. chemicals,
genes/proteins, diseases) extracted from the literature (there are several other NLP-derived
database that specialise in particular entity types and their relationships, for example the data-
base EVEX covers protein-protein interactions [13]). These are the 'Global Network of Bio-
medical Relationships' (GNBR) [14] and the 'Semantic MEDLINE' (SemMedDB) [15, 16].
Both GNBR and SemMedDB use a rules-based approach to extract information from sen-
tences; GNBR extracts a dependency path from each sentence using the 'Stanford Parser' [17],
while the 'Semantic MEDLINE' extracts subject-predicate-object triples from sentences using
an NLP engine called SemRep [16]. Within SemMedDB, predicates are normalised to a con-
trolled vocabulary (e.g. 'activates' OR 'agonises' will be normalised to STIMULATES), while
named entities are linked to the UMLS metathesaurus [18]. GNBR, however, uses a form of
distributional semantics [19, 20] to group similar sentences together. The authors developed
an 'Ensemble Biclustering' (EBC) algorithm [21] to group together similar sentences based on
their dependency path structure. With manual inspection of the groups, the authors were able
to derive categories, so that relationship themes were 'learned' from the literature itself.
Named entities were based on PubTators' co-occurrences dataset, which are normalised to rel-
evant ontologies (e.g., Medical Subject Headings (MeSH)).

The focus of our paper is an evaluation of a subset of the GNBR data; the relationships
between chemicals and genes/proteins. Within a drug-discovery context, it's more common to
describe small molecule-protein relationships rather than chemical-gene/protein relationships.
GNBR refers to chemicals-genes/proteins, presumably because the NER engine it relies on
identifies a broad range of chemicals (not just small-molecules) and because there is no differ-
entiation in entities that are referring to the gene or the protein in any given sentence (proteins

are normalised back to the gene ID using Entrez identifiers). Chemical-gene/protein relationships within GNBR fall into 10 categories, including 'inhibits', 'agonism' and 'antagonism'. For any given sentence, the EBC algorithm gives a score predicting how likely it is that this sentence belongs to each of the relationship classes. Most sentences are multi-labelled (i.e., they have a score in at least two categories) but generally have a much higher weight for one particular class.

Within the original GNBR paper, good agreement was shown between GNBR relationships and those from relevant databases when intersecting entity pairs were compared. However, the number of entity pairs compared was generally low. For example, the number of chemical-gene/protein entity pairs compared between GNBR and the Therapeutic Targets Database (now Open Targets) was just 195, 40 and 43 pairs for the classes 'Inhibition', 'Agonism' and 'Antagonism'.

We felt that further evaluation of the relationships scores were warranted against a larger dataset, especially for chemical-gene/protein pairs. Particularly, we were interested in evaluating the effect that different threshold values have on precision and recall, a topic that was not covered in the original paper. To do this, we have evaluated GNBR relationship scores against Evotec's federated database of interactions between small molecules and proteins, internally referred to as Nexus. The database is constructed from several public domain and commercial sources, including ChEMBL, DrugBank and Integrity/CDDI. Following the ChEMBL schema, protein "targets" can comprise both individual proteins as well as protein complexes and, where the precise interacting entity is unclear, groups of related proteins. The primary focus is on interactions that have an associated IC50, Ki or equivalent dose response characteristic. Where specified in the original data source, the type of interaction–for example, agonism, antagonism, inverse agonism—is also recorded. Currently, 57% of the 3.8 million individual (unaggregated) interactions have such a type associated with them.

We hypothesize that there should be many well-known chemical-protein interactions that are frequently mentioned in the literature and thus incorporated into GNBR that should intersect with Nexus. If GNBR relationship scores are to be trusted, there should also be a high degree of agreement between GNBR and Nexus relationships for interactions that are present in both sources. We were also interested in the difference between chemical-gene/protein entities in GNBR and Nexus. One major distinction between the two datasets is that Nexus deals predominantly with *direct* chemical-protein interactions, whereas GNBR also contains *indirect* chemical-gene or chemical-protein associations. For example, an author may mention that a certain chemical was associated with the upregulation of a gene, or that a chemical decreased the amount of a certain protein in a cellular assay. These indirect effects could be valuable additions to a knowledge graph for inference and prediction tasks, so long as the relationship can be trusted. Good agreement between GNBR and Nexus for experimentally quantified interactions will give more confidence for novel, NLP-derived interactions.

## Methods

All data manipulations and visualizations were performed in Python using Pandas, Seaborn and other relevant packages. The code is available in GitHub: https://github.com/Evotec-isRD/gnbr_nexus_comparison

We downloaded the latest version of GNBR from Zenodo https://zenodo.org/record/3459420#.YegqNBrP2Uk. We used the chemical-gene datasets: 'part-i-chemical-gene-path-theme-distributions.txt.gz', which contains the relationship weights per category for each sentence, and 'part-ii-dependency-paths-chemical-gene-sorted-with-themes.txt.gz' which contains entities, sentences, and dependency paths. As a first step, we combined these files into

one dataframe by mapping the dependency paths from 'part-i' and 'part-ii', so that each sentence and the associated relationship weights were all in one table.

The approximately 3.8 million individual interactions between compounds and protein targets in Nexus are aggregated where multiple values (e.g., from different scientific papers) exist for the same entity pair. In some cases, there are multiple different interaction types (e.g., 'inhibits' and 'binding') assigned for a given compound-protein pair. For the purposes of using Nexus as a ground truth to compare GNBR against, we filtered for compound-protein pairs that only had a single mode of action.

To find the intersection and differences between chemicals and gene/proteins in GNBR and Nexus, it was necessary to normalise names so that the entities could be matched between data sources. Nexus contains a list of compound synonyms as well as other identifiers such as ChEMBL and DrugBank IDs. GNBR includes the chemical name that was used in a sentence and, where this can be mapped, the corresponding MESH identifier. However, the proportion of chemicals that have a MESH ID assigned in GNBR is less than 35% (see results), so we decided to match strings between chemical names in GNBR and Nexus. However, we found there were multiple spellings of the same chemical within GNBR, which reflects the errors or variations in human writing. Therefore, exact string matching did not adequately capture all matches between Nexus and GNBR. For example, the chemical 'zinc-protoporphyrin IX' was spelt ten different ways in GNBR, from dropping the hyphen, to using '9' instead of 'IX', to including an extra 'e' at the end of the word. To mitigate against spelling differences, we used 'fuzzy matching', which calculates the Levenshtein distance between two strings (we used the Python package 'RapidFuzz'). We found that a distance threshold of 0.95 gave a good balance between identifying the most common differences (white space, hyphens, the addition/deletion/transposition of a single letter) between the same chemical and separating genuinely different chemicals. We were also careful to include some exceptions, so that common chemical substitutions would not be 'fuzzy-matched' (i.e. 'methyl' and 'ethyl', Z- and S-, '+' and '-', and 'E-' and 'R-'). A possible way to extend this analysis in the future would be to explore a machine-learning approach that can match chemical and gene concepts based on semantics rather than fuzzy syntactic structures.

Unlike chemical names, the multiple synonyms used for the same gene/protein in GNBR have already been normalised to NCBI ENTREZ gene identifiers (see PubTator's methods [8]), and there is good coverage (see results). PubTator's algorithm does not identify whether a gene or a protein is being mentioned in a sentence, hence all proteins are mapped to gene IDs. For proteins, Nexus uses UniProt accession numbers as identifiers. Therefore, we mapped GNBR ENTREZ gene IDs to UniProt protein IDs using UniProt's mapping functionality*, so that proteins/genes could be matched between Nexus and GNBR. *(https://ftp.uniprot.org/pub/databases/uniprot/current_release/knowledgebase/idmapping/).

For each sentence, GNBR provides a dependency path (S1 Fig in S1 File) with an associated score for each of the 10 relationship classes. These scores indicate how likely it is that the relationship identified in that sentence falls into each of the predefined classes. The Ensemble Biclustering algorithm [21] used in GNBR, scores sentence dependency paths relative to 'flagship' dependency paths for any given relationship. Most scores are in the range 1–100, but there are a small minority of sentences with scores up to 40,000, depending on the relationship class (S2 Fig in S1 File). To use such raw scores directly would lead to skewed selection towards specific relationships. Therefore, we normalised the scores by taking the fraction of score support per sentence across all ten relationship classes.

Below is a summary of the steps taken to preprocess GNBR data so that we could evaluate it against Nexus:

1. 'Fuzzy match' GNBR chemical names against Nexus chemical synonyms and IDs to create a dictionary of GNBR-Nexus chemical matches accounting for slight misspelling, hyphenations etc, and multiple names used for the same entity within GNBR.

2. Map GNBR gene ENTREZ IDs to UniProt accession numbers in Nexus.

3. Normalise GNBR scores by taking the fraction of score support per sentence (i.e., per row in the dataset) across all ten relationship classes.

After preprocessing the GNBR data for compatibility with Nexus, we merged the two datasets to evaluate GNBR's relationship scores. The following is a summary of the steps taken to do this:

1. We merged GNBR and Nexus on the 'fuzzy-matched' chemical names and UniProt protein IDs for the intersection of chemical-protein pairs that are present in both datasets. This results in 157,643 rows from GNBR (from an original total size 1.74M rows), of which there are 9990 unique chemical-protein pairs.

2. We chose four relationship themes in GNBR that map directly to Nexus relationship categories for comparison. These are: '(N) inhibits', '(A+) agonism', '(B) binding', '(A-) antagonism'. Note, we have shortened some of the original GNBR classes for brevity, for instance, '(B) binding, especially to ligands' is now simply, '(B) binding' (For a full list of GNBR classes see S2 Fig in S1 File).

We evaluate Receiver Operating Characteristic (ROC) and Precision Recall Curves (PRC), using the Nexus relationships as the true classes.

## Results

### Comparison of chemicals and genes/proteins found in GNBR and Nexus

Fig 1 shows the number of unique chemical and gene/protein names in GNBR. There are 64,672 unique chemical names but only 22,863 unique MESH identifiers. In contrast, there are 58,229 unique gene/protein names with 47,118 unique ENTREZ gene identifiers. The reason that gene/protein have a higher proportion of mappings between raw text and ENTREZ IDs than chemical names in text to MESH IDs is likely due to completeness in the knowledge bases. MESH does contain IDs for common chemicals and drugs, particularly those with known therapeutic applications (e.g. aspirin), but not for many other compounds. Other specialized chemical knowledge bases such as PubChem, ChEMBL or Nexus offer a wider coverage of chemical names and synonyms.

Fig 2 shows Venn diagrams summarising the chemical and gene/protein entities in both Nexus and GNBR. A few pertinent observations can be made regarding the differences and intersections of the two datasets:

i. Nexus has about 17 times more unique chemicals than GNBR. There are a number of possible reasons for this. For example, often chemicals are referred to as part of a series like "Compound 19C" when they are synthesized with differing substitutions (e.g. ethyl for methyl), with the actual chemical details embedded in a table or figure. Moreover, only around 28% of articles are open access and thus GNBR/PubTator algorithms have no access to the full body of the text, while curators of databases like ChEMBL (which Nexus incorporates) often do. Another reason why NLP-derived data can be limited is because research generally focuses on novel relationships. Thus, publications only reference a few of the vast corpora of known biomedical relationships in structured sources like ChEMBL. Further, many of the 'chemicals' that are unique to GNBR fall into categories such as

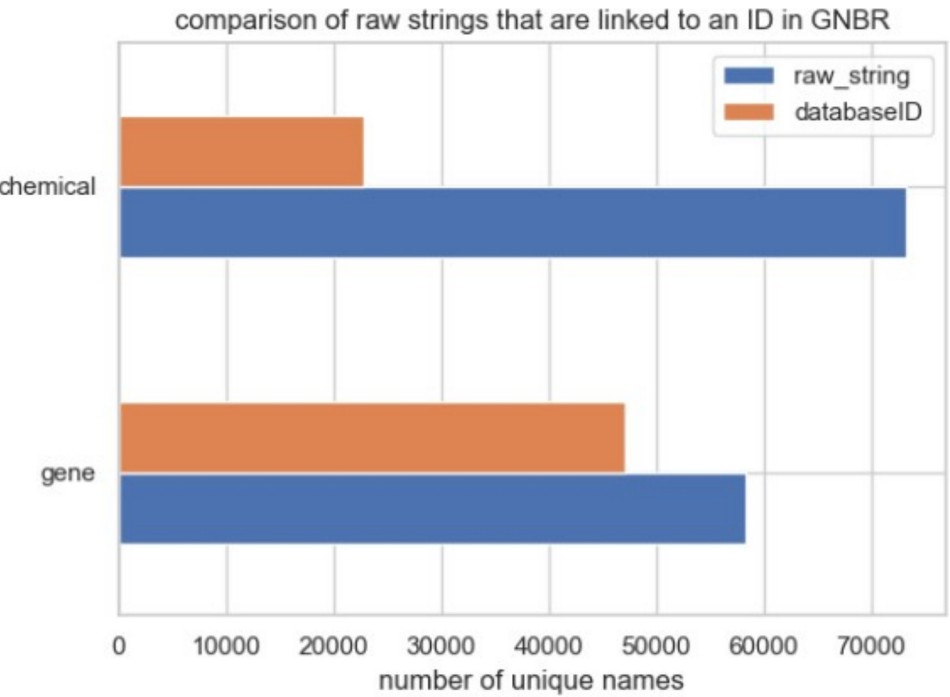

comparison of raw strings that are linked to an ID in GNBR

**Fig 1. Comparison of raw strings that are linked to a database identifier in GNBR.** Most of the chemicals do not have an identifier in contrast to the gene/proteins.

'alcohols' (e.g. ethanol), 'detergents' (e.g. triton-X), 'elements' (e.g. calcium, iron, etc), and 'metabolites' (e.g. fatty acids, carbohydrates, amino acids, nucleotides). In fact, many of these entities are in Nexus, but tend to have a more specific nomenclature, such as 'calcium salt' rather than just 'calcium'. Here, the limitations of chemical name fuzzy matching between Nexus and GNBR are apparent. In addition, GNBR does include large molecules whereas Nexus has a molecular weight cutoff threshold of 1200. For instance, Cyclosporin A has a molecular weight (MW) of 1202.6, and so appears in GNBR but not in Nexus. Chemicals at the intersection of both datasets tend to be drugs, such as 'aspirin', or commonly used chemicals. A further discussion of intersecting chemicals is in point (iii).

ii. GNBR has a higher proportion of non-human gene/protein mentions than Nexus. This is unsurprising as PubTator's NLP NER recognises all gene/protein mentions regardless of species, whereas in Nexus, the data is more focused towards collecting data on human protein targets (or commonly used surrogate species such as mice, rat, or monkey). The genes/proteins that are unique to GNBR range from 'Bovine' through to 'Xenopus' (frog). The intersection of ~4K proteins in GNBR and Nexus are mostly human.

iii. Nexus has around 17 times more unique chemical-protein pairs than GNBR. This can be largely attributed to the increased number of unique chemicals that are in Nexus. The similarities and differences between the chemical-protein pairs that are unique or different between the two datasets are highlighted with a case study in the following section.

Table 1 takes aspirin as a case study to highlight the similarities and differences between GNBR and Nexus. Both datasets have the very well-known inhibiting interaction of aspirin with Prostaglandin G/H synthase 1 (also commonly known as cyclooxygenase-1 [COX-1]).

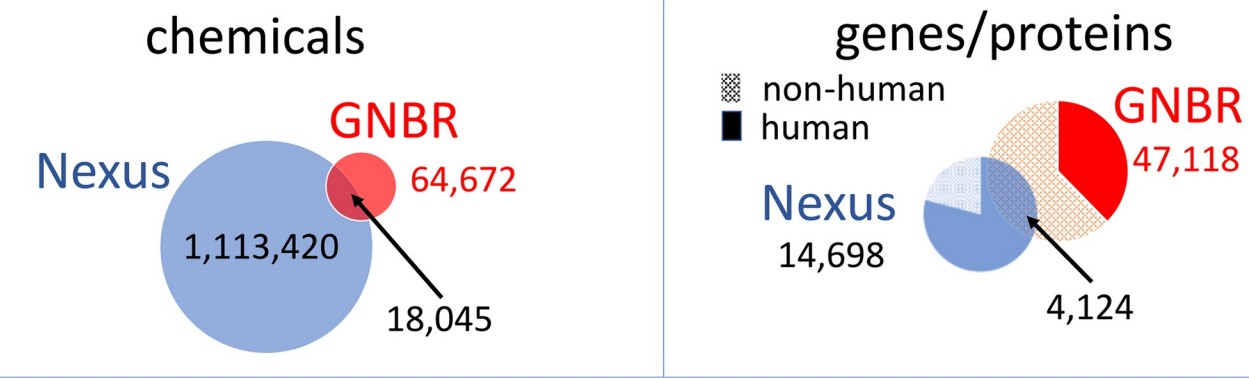

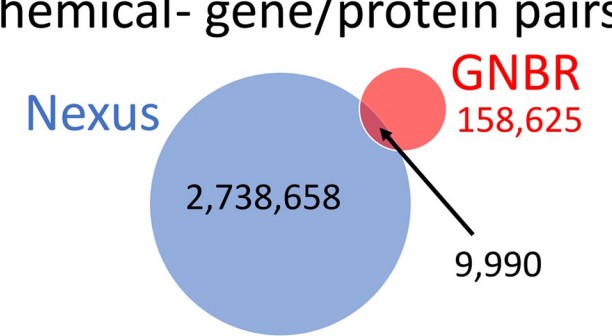

**Fig 2. Venn diagrams showing the intersection and differences between "chemicals", "genes/proteins" and "chemical-genes/proteins pairs" between GNBR and Nexus.** Here, we are using a subset of Nexus where chemical-protein pairs have been aggregated from 3.8 individual measurements extracted from various sources (e.g. patents, scientific papers).

This inhibition has the downstream consequences of pain relief, as COX-1 produces prostaglandins that, amongst other functions, are pain messengers.

Interestingly, COX-1 highlights the pitfalls of named entity recognition and entity linking which is a source of error in GNBR. "COX-1" is an abbreviation for two distinct proteins: (1) Prostaglandin G/H synthase 1 [UniProtID: P23219] and (2) 'cytochrome c oxidase' [UniProtID P00395]. Within GNBR, all instances of 'COX-1' that are referring to Prostaglandin G/H synthase are mis-labelled as 'cytochrome c oxidase'. Here, Named Entity Recognition has performed correctly in assigning COX-1 a protein label. But, the subsequent step after NER of Entity Resolution, when an entity type (e.g. protein in this case) is then mapped to a concept ID in a known ontology (e.g. UniProt) is incorrect. Thus, in GNBR, aspirin is wrongly associated with 'cytochrome c oxidase', and consequently there are fewer instances of 'aspirin-cyclooxygenase1'. GNBR uses PubTator's named entity recognition from 2019, but a recent search

**Table 1. Using aspirin as an example drug, we show three examples of a protein target that are unique to either GNBR or Nexus, or common to both.**

| Dataset | Chemical | Protein | An example dependency path from a sentence in GNBR | Relationship in Nexus |
|---|---|---|---|---|
| GNBR | aspirin | thrombin | inhibited\|nmod\|START_ENTITY inhibited\|nsubjpass\|response response\|nmod\| END_ENTITY | Not in Nexus |
| NEXUS | aspirin | interleukin 1 receptor type 1 | Not in GNBR | Antagonist |
| GNBR and NEXUS | aspirin | cyclooxygenase-1 | START_ENTITY \|nmod\|agent agent\|nmod\|inactivation inactivation\|nmod\| END_ENTITY | Inhibitor |

by us (July 2022) showed that 'COX-1' is still normalised to the wrong gene 'cytochrome c oxidase' in PMID35768902. So as far as we are aware the problem persists. Of note, the Open Targets co-occurrence dataset correctly identifies COX-1 as cyclooxoygenase-1 in sentences involving aspirin. This is probably due to a bioBERT transformer model used for NER, which can distinguish COX1 as either 'cyclooxoygenase-1' or 'cytochrome c oxidase' depending on the context (Open Targets blog: https://blog.opentargets.org/developing-the-open-targets-literature-pipeline/)

One association that is unique to Nexus is that aspirin is an antagonist of Interleukin-1 receptor type 1. The source of this information is a single pay-walled article [22] where the association is not explicitly mentioned in the abstract, and thus it is hardly surprising that it does not in appear in GNBR which uses MEDLINE abstracts as its source material.

A unique aspirin-protein association that appears in GNBR but not Nexus is with thrombin. GNBR has 161 separate sentences that mention aspirin and thrombin, with 'inhibits' occurring frequently in the dependency path (where the verb is often the root of the sentence). Aspirin is an anticoagulant and is used to treat some cardiovascular diseases. The therapeutic mechanism is not entirely understood, but aspirin could indirectly inhibit thrombin by directly inhibiting COX-1. COX-1 produces thromboxane A-2, which is needed by thrombin in the pathway that leads to platelet aggregation. Of course, it is not within the scope of Nexus (which is primarily concerned with direct interactions) to include such indirect effects, so it is not surprising that the association is not included. Interestingly, we performed a search for an association of aspirin to thrombin in a number of databases (Signor, KEGG, Reactome, STITCH, Open Targets, DrugBank, ChEMBL) and did not find direct reference to those two entities together (although STITCH associates aspirin to thromboxane A-2 via co-occurrence in a PubMed abstract). Although not exhaustive, this suggests that this entity pair is unique to GNBR/PubTator, and the triple is unique to GNBR. Here then is an example of a triple that is currently only found in an NLP-derived dataset.

## Evaluation of the relationship classification predictions from GNBR

In Fig 3 and Table 2, we show some randomly selected rows from the intersection of GNBR and Nexus to help illustrate how the GNBR relationship scores compare to Nexus categories. For instance, it can be seen in Fig 3 that, within GNBR, the chemical 'Dicumerol' and the protein 'Quinone Reductase 1' have the highest score for the class '(N) inhibits' of 0.75, while also having a relatively high score for '(E-) reduces expression or production' of 0.12. In Nexus, the chemical 'Dicumerol' is an 'inhibitor' of 'Quinone Reductase 1'. With this example, a threshold value equal to or more than 0.75 would result in a matching 'inhibits' relationship between GNBR and Nexus.

Table 2 shows the sentence and dependency path from which the examples shown in Table 2 are derived. For the 'Dicumerol-Quinone Reductase 1' chemical-protein pair, note that the pertinent part of the sentence is '. . .addition of the DTD inhibitor, dicumarol. . ..' (DTD is a synonym of 'Quinone Reductase 1'), with the dependency path picking out the verbs, 'addition' and 'inhibitor'.

As an example of how GNBR weights relationship classes for a particular chemical-protein pair, we can again consider 'aspirin-cyclooxygenase1'. Fig 4 shows an upset plot to visualise the relationships assigned to 442 sentences where 'aspirin' and 'cyclooxygenase-1' are the chemical and protein entities, with a threshold set at 0.5 on the EBC score. Here, 280 sentences are exclusively true for '(N) inhibits'; 2 sentences are true for either '(K) metabolism/pharmacokinetics' or '(E) affects production'; 2 sentences are true for both '(N) inhibits' and '(O) transport'; 2 sentences are true for both '(N) inhibits' and '(K) metabolism/pharmacokinetics';

| Index | chemical, raw string | Protein name | (N) inhibits | (A+) agonism | (B) binding | (A-) antagonism | (E+) incr. expr/prod | (E-) decr. expr/prod | (E) affects prod. | (O) transport | (K) Metabol / pharmk | (Z) enzyme | Nexus |
|---|---|---|---|---|---|---|---|---|---|---|---|---|---|
| 3172 | dicumarol | Quinone reductase 1 (DTD) | 0.72 | 0 | 0 | 0.05 | 0 | 0.12 | 0.07 | 0 | 0.02 | 0.02 | Inhibitor |
| 4172 | fut-175 | Complement C1s | 0.01 | 0.01 | 0.76 | 0.02 | 0.01 | 0.01 | 0.05 | 0.1 | 0.01 | 0.02 | Inhibitor |
| 4430 | 10058-f4 | Myc proto-oncogene protein (c-myc) | 0.1 | 0 | 0.1 | 0 | 0 | 0.2 | 0.5 | 0 | 0.1 | 0 | Inhibitor |
| 5594 | lenalidomide | TNF-alpha | 0.33 | 0 | 0 | 0 | 0 | 0.67 | 0 | 0 | 0 | 0 | Inhibitor |
| 6010 | mdl100907 | Serotonin (5-HT) receptor | 0.13 | 0.02 | 0.04 | 0.57 | 0 | 0.06 | 0.15 | 0 | 0.04 | 0 | Antagonist |
| 6157 | metolazone | Pregnane X receptor | 0 | 0 | 0 | 0 | 0.5 | 0 | 0.5 | 0 | 0 | 0 | Agonist |
| 6331 | ml174 | Protein phosphatase methylesterase 1 | 0.5 | 0 | 0 | 0 | 0 | 0 | 0 | 0 | 0.5 | 0 | Inhibitor |
| 6472 | mrs2365 | Purinergic receptor P2Y1 | 0.25 | 0.5 | 0 | 0.25 | 0 | 0 | 0 | 0 | 0 | 0 | Agonist |
| 6535 | n-3-benzyl-phenobarbital | Cytochrome P450 2C19 | 0.5 | 0 | 0 | 0 | 0 | 0 | 0.5 | 0 | 0 | 0 | Inhibitor |
| 9433 | succinate | Succinate dehydrogenase | 0 | 0 | 0.02 | 0 | 0 | 0 | 0.01 | 0.05 | 0.01 | 0.9 | Binding |

**Fig 3. The intersection of GNBR and Nexus based on chemical-protein pairs, consists of 157,643 rows, where each row contains a chemical-protein and the GNBR relationship score for each class between them, all of which are derived from a single sentence extracted from an article in PubMed.** The sentences and dependency paths are shown in Table 2, related by the 'Index' column. The relationship found in Nexus for a particular chemical-protein pair is added to each row and is treated as ground-truth. The cells in red indicate the highest score across the GNBR relationship classes, while the ones in yellow are the second highest.

while 154 sentences do not have enough weight for any relationship category at the threshold value. S3 Fig in S1 File shows an upset plot with no threshold. Here, only 54 sentences are exclusively true for '(N) inhibits', while 206 sentences are true in four categories ('(N) inhibits', '(E) affect production', '(E-) decreases expression/production', '(K) metabolism/

**Table 2. The sentence and dependency path for the chemical-protein entities shown in Fig 3.** The entities for each sentence are highlighted in bold.

| Index | PMID | Sentence, tokenized | Dependency path |
|---|---|---|---|
| 3172 | 15655414 | addition of the **DTD** inhibitor, **dicumarol**, significantly inhibited cytotoxicity of mmc and sn, and reversed the increased cytotoxicity seen when dmf was combined with either mmc or sn in all glioblastoma cell lines | addition\|appos\|start_entity addition\|nmod\| inhibitor inhibitor\|compound\|end_entity |
| 4172 | 27810412 | positive therapeutic effects in ami animal models have been described for cobra venom factor, **soluble complement receptor 1**, c1-esterase inhibitor (c1-inh), **fut-175**, c1s-inhibitor, anti-c5, adc-1004, clusterin, and glycosaminoglycans. | receptor\|amod\|start_entity receptor\|amod\| end_entity |
| 4430 | 27789709 | here, we show that the c-myc inhibitor **10058-f4** blocks the induction of **c-myc**, prc, and representative prc-dependent stress genes by the respiratory chain uncoupler, carbonyl_cyanide_m-chlorophenyl_hydrazine (CCCP). | blocks\|nsubj\|start_entity blocks\|dobj\|induction induction\|nmod\|end_entity |
| 5594 | 22997574 | this pilot study evaluated **lenalidomide** at reduction of **TNF-a** and improvement of behavior and language in children with autism with elevated TNF-a. | evaluated\|dobj\|start_entity evaluated\|nmod\| reduction reduction\|nmod\|end_entity |
| 6010 | 12585687 | in contrast, sb-200646 failed to modify the facilitatory procognitive effect produced by (+ /—)-2.5-dimethoxy-4-iodoamphetamine (DOI) or ketanserin, which were sensitive to **mdl100907** (a selective **5-HT2a receptor** antagonist) and to a ly215840 high dose. | start_entity\|appos\|antagonist antagonist\|amod\| end_entity |
| 6157 | 25181459 | to determine the role of fda-approved drugs in pxr-mediated regulation of drug metabolism and clearance, we screened 1481 fda-approved small-molecule drugs by using a luciferase reporter assay in hek293t cells and identified the diuretic drug **metolazone** as an activator of **hPRX**. | identified\|dobj\|start_entity identified\|nmod\| activator activator\|nmod\|end_entity |
| 6331 | 22834039 | the scripps research institute molecular screening center -lrb- srimsc -rrb-, part of the molecular libraries probe production centers network (mlpcn), identified a potent and selective **PME-1** inhibitor probe, **ml174**, by high-throughput screening using fluorescence polarization-activity-based protein profiling (fluopol-abpp). | probe\|appos\|start_entity probe\|compound\| end_entity |
| 6472 | 25327170 | addition of the **p2y1** agonist **mrs2365** 10 m during the purinergic rundown did not cause any hyperpolarization. | m\|compound\|start_entity m\|compound\| end_entity |
| 6535 | 11854139 | therefore, (+)-n-3-benzyl-nirvanol and (-)-**n-3-benzyl-phenobarbital** represent new, highly potent and selective inhibitors of **cyp2c19** that are likely to prove generally useful for screening purposes during early phases of drug metabolism studies with new chemical entities. | represent\|nsubj\|start_entity represent\|dobj\| inhibitors inhibitors\|nmod\|end_entity |
| 9433 | 22152483 | we have identified a further subunit of the carrier translocase (tim22 complex) that surprisingly is identical to subunit 3 of respiratory complex ii, **succinate** dehydrogenase **(SDH3).** | dehydrogenase\|amod\|start_entity dehydrogenase\| appos\|end_entity |

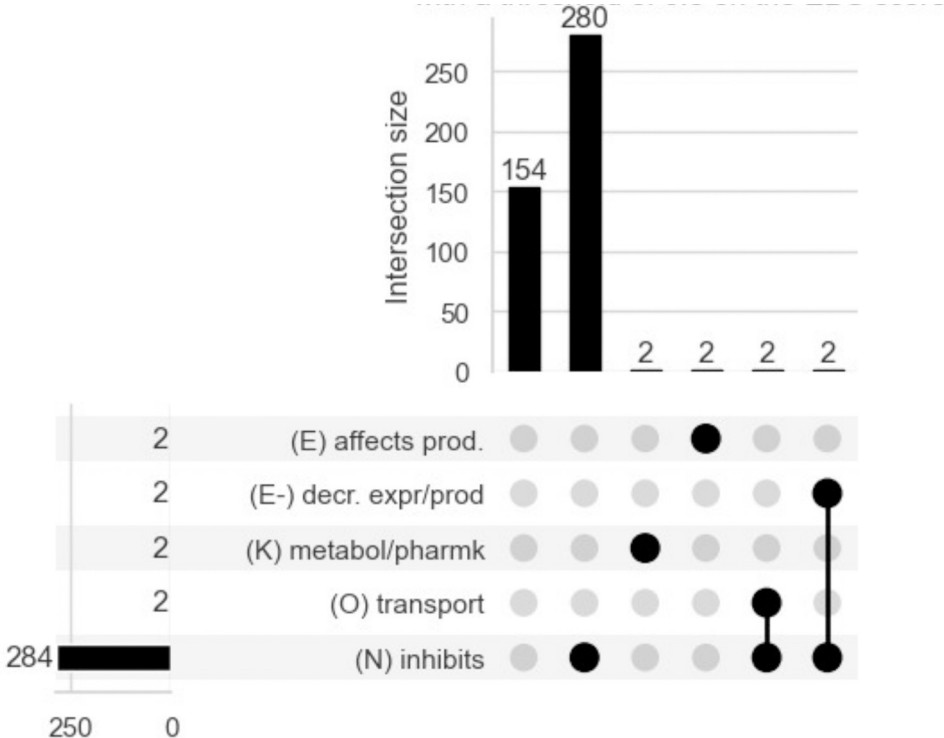

**Fig 4. An upset plot showing the relationships assigned to 442 sentences where 'aspirin' and 'cyclooxygenase-1'
are the chemical and protein entities, with an EBC score threshold set at >0.5.** An upset plot can be read as follows:
the horizontal left-hand bars show the total number of sentences that had any given category as true. For example, 284
sentences had '(N) inhibits' as true. The vertical top-most bars show sentences that have the categories indicated with a
black dot as true. Here, 154 sentences have no black dot, which indicates they were false in all categories (i.e. all values
were lower than the 0.5 threshold); 280 sentences are true for '(N) inhibits'; 2 sentences are true for either '(K)
metabolism/pharmacokinetics' or '(E) affects production'; 2 sentences are true for both '(N) inhibits' and '(O)
transport); while 2 sentences are true for both '(N) inhibits' and '(K) metabolism/pharmacokinetics'.

pharmacokinetics'. With the higher threshold at 0.5, more than half the sentences have been
classified correctly in comparison to Nexus.

Fig 5A shows the balance of Nexus labels that can be evaluated against GNBR sentences
(i.e., at the intersection between the two databases) as well as the total number of sentences
(rows) at the intersection of the two datasets. There are about four times more chemical-gene
pairs that have 'inhibitor' as a relationship compared to 'agonist' and 'antagonist', while 'bind-
ing' has the fewest instances. While the class 'inhibits' is relatively well balanced compared to
the total observation in the dataset, with just under half of the chemical-protein pairs having
this relationship, the other classes are unbalanced, with each class having less than 13% of the
total observations.

Fig 5B shows the ROC curves comparing GNBR relationship scores to Nexus labels as
ground truth, performed on the whole dataset of 157,643 rows. Of the 4 relationship classes in
GNBR, 'inhibits' is the best predicted, while 'binding' is the worst. The ROC AUC values for
'inhibits', 'agonism', 'antagonism' and 'binding' are 0.73, 0.71, 0.63 and 0.50, respectively. It is
worth noting that ROC curves are relatively sensitive to class imbalance compared to precision
recall curves [23]. This is because PRCs highlight the number of false positives relative to the
class size, whereas ROC curves better reflect the total number of false positives independent of
which class (positive or negative) they come from.

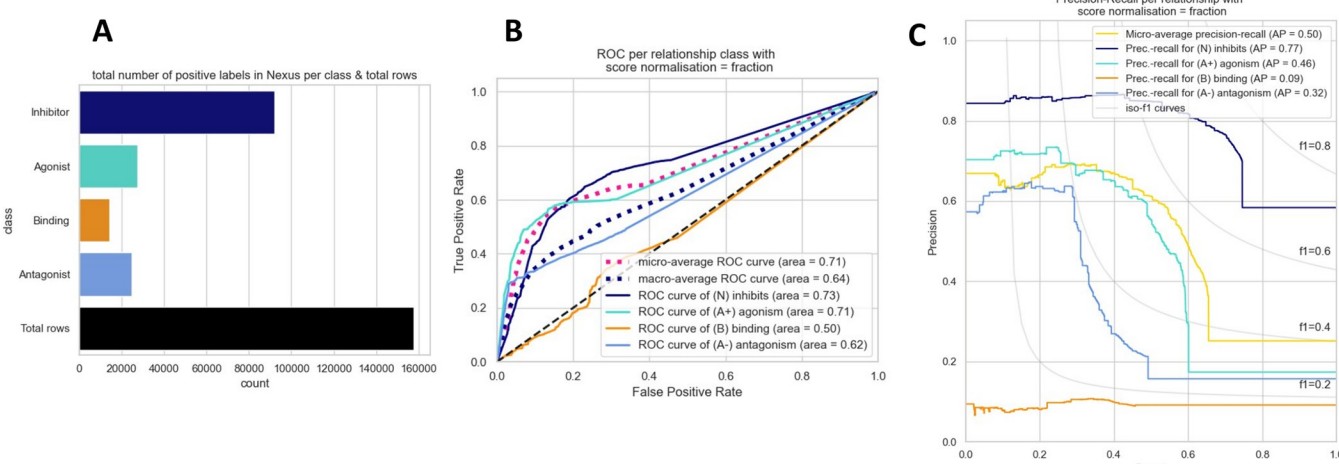

**Fig 5. A.** The total number of positive labels in Nexus per relationship category, and the total number of sentences (rows) in the dataset that intersect with GNBR. It is clear there is a large class imbalance with relationships 'Agonist', 'Binding' and 'Antagonist', while 'Inhibitor' is more than half of the total observations. **B.** Receiver Operator Curves (ROC) and **C.** Precision-Recall curves (PRC) comparing GNBR to Nexus as ground-truth.

Fig 5C shows the PRC AUC, which go in the same order as shown by the ROC curves —'inhibits', 'agonism', 'antagonism' and 'binding'—with 'inhibits' having the highest AUC value of 0.77 and 'binding' the lowest of 0.09. However, with the ROC curves the 'inhibits and 'agonist' class prediction performed similarly well, while with the PRCs we see that 'agonism' is considerably worse at nearly half that of 'inhibits' (0.77 and 0.44 for 'inhibits' and 'agonist', respectively). Further, GNBR 'binding' prediction has a very low PRC AUC of 0.09.

To try to understand why the predictive performance is better on certain classes than others, we can examine the examples in Fig 3 & Table 2. There are six examples of chemicals that Nexus classed as 'inhibitors' of the respective protein (row index: 3172, 6331, 6535, 4172, 5594, 4330), two examples of 'agonists', one of 'antagonists' and one of 'binding'.

From the examples of Nexus' 'inhibitor', three have relatively high GNBR scores of > = 0.5 in the GNBR '(N) inhibits' category (index: 3172, 6331, 6535). Here, within the sentence the word 'inhibits' appears (and is picked out in the dependency path for index 3172 and 6331), so intuitively it is easy to see how this is classed in the '(N) inhibits' category. Row 4172 has the highest score in '(B) binding', and the dependency path has picked out the word 'receptor'. This makes sense as 'binding' and 'receptor' are two associated words. However, this relationship is false in the context of the sentence, which is just a list of entities. This highlights one area of error in GNBR: a relationship is assumed to exist between two entities when in fact they just cooccur within the same sentence. GNBR's use of cooccurrence likely generates significant false positives as many sentences have cooccurring entities that have no semantic relationship. Modern NLP approaches will mitigate this challenge [11]. Row 4330 and 5594 exemplify how authors can express the consequence of 'inhibition', rather than stating it explicitly. Row 4330 has the highest score in '(E) affects expression', presumably because 'block' and 'induction' appear in the dependency path from the phrase "here, we show that the c-myc inhibitor 10058-f4 blocks the induction of c-myc". In the context of the sentence, GNBR is correct in categorizing as 'affecting expression' as the authors have used the word 'induction' which implies genetic transcription. Here there may be a conflation between the actual mechanism of 10058-f4 and its effect. Its effect is that the protein c-myc does not have so much influence in the cell which can occur either because there are fewer proteins due to transcription blocking, or because the proteins themselves are inhibited. In fact, 10058-f4 specifically inhibits c-Myc-

Max interaction and prevents transactivation of c-Myc target gene expression, so GNBR is correct here. Similarly, in row 5594 the phrase "evaluated lenalidomide at reduction of TNF-a" likely results in the highest score for '(E-) decreases expression or production', whereas Nexus classifies this relationship as 'Inhibits'. Here we assume the authors were referring to a reduction in the activity of TNF-a as a result of its inhibition by lenalidomide, so to categorise the relationship as decreasing activity (production) would be correct. Further, GNBR also gives this sentence 0.33 in the '(N) inhibits' class, likely linking the word 'reduction' to the class 'inhibits'.

For the 'agonist' examples, one sentence agrees with Nexus, while the other has joint high scores in the '(E+) increases expression/production' and '(E) affects production' category. For the 'antagonist' example, GNBR has the highest score in '(A-) antagonism', and so agrees with Nexus in this case. The example that Nexus classifies as 'Binding', is succinate to succinate dehydrogenase. Within the dependency path of this sentence is 'dehydrogenase', which probably contributes significantly to its high score in '(Z) enzymes'. In the context of the sentence, classifying the relation as enzymatic does make sense–but does not agree with Nexus' label.

Fig 6A–6D visualises the percentage of sentences that fell into GNBR's ten categories after applying a high threshold of 0.9. The aim here is to determine if, by applying a high threshold, a sub-set of high-fidelity triples can be extracted from GNBR that have very high agreement with Nexus.

Fig 6A shows chemical-protein interactions from Nexus' 'inhibitors' category that intersect with GNBR. With this high threshold, 77.9% of the sentences do not have high enough predictive scores for any relationship category. After those are excluded, most sentences are predicted to be in one of the relationships '(N) inhibits', '(E) affects production', '(E-) decreases expression or production' or '(K) metabolism/pharmacokinetics', in that order of frequency. For Nexus' 'Agonist' class, the two top predictions in GNBR are '(B) binding' and '(A+) agonism'. However, a significant proportion of sentences are predicted as '(N) inhibits' or '(A-) antagonists', which conceptually are the opposite of agonists although there may be issues with respect to the correct labeling of inverse (ant)agonists. For Nexus' 'Antagonist' class, GNBR predicts most of these sentences as '(N) inhibits', '(B) binding', '(A-) antagonism' and '(A+) agonism'. We have noted that sentences do often get relatively high scores in both '(A-) antagonism' and '(A+) agonism' that, although these are mutually exclusive concepts, probably reflects the similarities in the dependency path structure of the sentences.

GNBR is poor at predicting 'Binding' labels from Nexus. 'Binding' is a concept that is true for all direct chemical-protein interaction, as to exert an effect the chemical and the protein must come into contact. In Nexus, the term tends to be used when the functional effect such as inhibition of the chemical binding to the protein is not known, but the interaction has been verified by some experimental method (e.g. crystallography). Here then, GNBR's prediction against Nexus' 'Binding' category is not necessarily wrong, rather GNBR could be adding new information to a particular chemical-protein pair that relates to the effect or nature of the binding–as in the 'dehydrogenase' example mentioned earlier.

We also evaluated the percentage of sentences that had multi-labels with a lower threshold. We found that only a very low percentage of sentences had multi-labels at a threshold of 0.5, with none being classified with more than two separate classes (see S4 Fig in S1 File).

Overall, Fig 6 shows that applying a threshold will not give subsets of sentences that have high agreement with Nexus. Consequently, we argue that filtering GNBR data for what might be described as 'high-fidelity triples' is not a simple task.

## Discussion

The main motivation for this study was to evaluate the validity of relationship types assigned to chemical-gene/protein pairs within GNBR and to assess implications for incorporation of

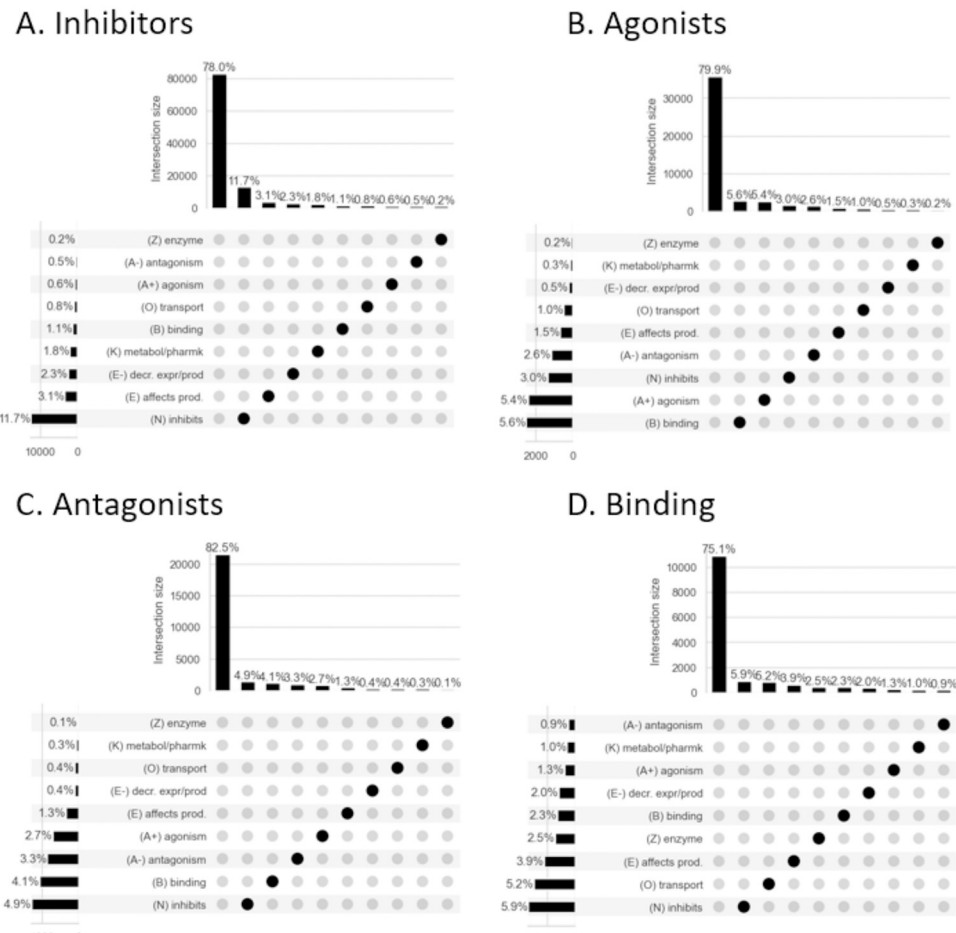

**Fig 6.** A visualization of how GNBR categorises chemical-protein relationships compared to Nexus' four classes (A-D) at a threshold of 0.9. We plot the percentage of sentences that fall into the GNBR relationship themes or do not have a high enough score for any category–which at this high threshold is most sentences (~75–80%). The only category that has the highest percentage of sentences in agreement with Nexus are 'Inhibitors' (11.7%). At lower thresholds sentences can have equal weights in >1 category (see S3, S4 Figs in S1 File as examples).

this data into biomedical knowledge graphs. Overall, GNBR and Nexus had reasonable agreement for the classes of 'inhibitor' and 'agonist', with poorer agreement for 'antagonist' and 'binding'.

We decided to compare individual sentences within GNBR to Nexus, as we wanted to assess the variability in classification for any given chemical-protein relationship. However, if we had aggregated the mentions of any chemical-protein pair together and retained only the most frequent relationship, it is likely that the agreement between GNBR and Nexus would be improved. For instance, as shown with our case study of aspirin, at a threshold of 0.5 most of the sentences were exclusively categorised as 'inhibits' in agreement with Nexus. That said, many chemical-protein pairs within GNBR are only mentioned in a few sentences, or sometimes only once, in which case an aggregated ranking scheme would make less of a difference. Furthermore, novel associations with infrequent mentions, can be important, especially if they are not in structured databases.

In contrast to single label definitions, GNBR's weighted scores across several categories could give a more nuanced representation of entity relationships. This is probably not applicable in the context of the current analysis, where a chemical-protein has a very defined, direct relationship (e.g. agonist or antagonist or inhibitor—arguably 'binding' is less defined) and it does not make sense to attribute other relationships like 'increases production'. The exception to this is where a gene is mentioned in a sentence–as opposed to the protein–in which case some of the GNBR relationships such as 'increases production' could be more relevant than an arguably more chemical-protein centric relationship such as 'agonist'. Further, GNBR's relationships could be entirely appropriate where the entities are more complex, such as diseases or cell processes, and the relationships between them are multi-faceted and could fall into a number of different complementary categories.

In the context of knowledge graphs, GNBR has been used as the base data where the relationships weights have either been utilised or ignored. For example, Zeng *et al.* used a knowledge graph embedding and link prediction algorithm, RotateE, to embed the entities and relationships contained within GNBR to predict 41 repurposable drugs to treat COVID-19 [24]. These authors ignored relationship weights, presumably assigning a relationship class between entities if the score was more than zero. As we have shown in the current analysis, using no threshold on the relationship weights will result in poor precision, and consequently will result in many erroneous relationships. How important these errors are within a knowledge graph for accurate predictions is an interesting area of research, but one might reasonably assume has a detrimental effect on truthful inference.

In contrast, Sosa *et al.* took advantage of GNBRs' relationship weights in drug repurposing prediction for rare diseases [2]. These authors adopted an uncertain knowledge graph embedding method [25], which uses the support scores for each of the themes. In an embedding-based link prediction task they showed a high PRC AUC of 0.91 against a gold-standard dataset (MEDI), which comprises 811 drugs, 360 diseases and 3,329 associations of the two [26]. From this study, it seems that embedding GNBR data gives very high predictive performance. It would be interesting to perform a sentence-by-sentence comparison with MEDI, which would be comparable to the approach we have taken.

It is also interesting to compare results from knowledge graphs that are constructed using other NLP-derived datasets. Zhang *et al.* used the Semantic MEDLINE database (SemMedDB) for their knowledge graph and predicted 33 repurposable drugs to treat COVID-19 [3]. Interestingly, only one drug—estradiol–is common to both sets when Zeng *et al.*'s (using GNBR data) and Zhang *et al.*'s (using SemMedDB) predictions are compared. Perhaps this difference highlights the fragility of relationship categories and assignment, which will, in turn, have consequences for machine learning embeddings and prediction. For example, SemMedDB extracts 'subject, predicate, object' triples from sentences and then normalises the predicates to the UMLS Semantic Network ontology. This has some overlap with GNBR's relationship classes but also many differences. Further, Zhang *et al.* filtered information from an original, much larger knowledge graph to create a 'finer-grained' dataset more specific to COVID-19. This sort of data selection also affects predictions, and it is unclear how the predicative power was augmented by the various datasets.

GNBR uses a form of distributional semantics to learn themes pertaining to relationship classes from the scientific literature in a semi-supervised manner. The basis of these themes is clustering of similar sentences from their dependency path structure. However, there are several limitations with dependency parsing, not least that a large proportion of sentences were discarded from GNBR as they could not be assigned to any theme. With the release of BERT-style language models, it's now possible to implement topic modelling using BERT-embedded sentences [27]. This approach could capture meaning to a greater degree than dependency

parsing, and possibly allow for better disambiguation of entities with the same/similar names depending on context. It would be interesting to implement topic modelling using bioBERT and compare the results with the themes generated in GNBR. It is possible that a finer-grained relationship structure may be found, and coverage could be substantially increased due to fewer sentences being discarded. Other approaches such as Tellic's "Semantic Relationship Quantification" model–which scores the confidence in a meaningful sematic relationship between two entities in a sentence–could be used to check the accuracy of relationships in individual sentences [11].

It is also important to note that GNBR does not explicitly include any factuality labels for its triples. This contrasts with databases such as SemMedDB, which includes levels of factuality derived from the meaning of sentences, such as hypotheses, speculations, or opinions. These are on a scale from *Fact* to *Uncertain (Doubtful*, *Possible*, *Probable)* to *Counter Fact (e.g. drug does <u>not</u> treat disease)* [28]. However, it is possible that the factuality of a triple is implicitly encoded in the GNBR weights given to the relationship categories. GNBR's EBC clustering algorithm scores sentence dependency paths relative to 'flagship' dependency paths for any given relationship. If these 'flagship' dependency paths are based on sentences where a fact is stated, then it is likely that dependency paths that are very similar will also contain factual statements. Some investigation of the factuality of 'flagship' paths may be warranted here.

Depending on the level of error that one is willing to accept regarding the fidelity of relationships between entities, GNBR may (or may not) be regarded as a good base for a biomedical knowledge graph. However, for those with very high precision requirements, GNBR could certainly serve as an excellent resource to aid manual annotation of sentences. For example, BERT-style language models are good at classification tasks but perform better with larger and more varied training datasets [29, 30]. Such training sets for relationships between biological entities are in short supply and, arguably, do not fully represent the diversity found in the scientific literature [31]. An envisaged workflow could present human annotators with a selection of sentences from GNBR with entities highlighted and GNBR relationship weights displayed. The concentration required by an annotator to either agree with GNBR's classification or adjust accordingly would be greatly reduced compared to annotating with no aids. This could improve overall accuracy and provide larger training sets. It would probably also highlight some gaps in the GNBR relationship classes, which could then be expanded.

## Conclusion

We conclude that GNBR certainly contains a collection of unique chemical-protein associations with believable relationships, not found in any other structured databases like ChEMBL. These associations are likely to contain many indirect relationships which could be useful in drug discovery pipelines. Further, we discussed how GNBR is used as the basis for knowledge graphs for drug discovery but probably contains many erroneous associations which may make drawing conclusions fragile. This might be mitigated somewhat by using embedding techniques that consider weights and uncertainty. Finally, we foresee that GNBR could be very useful in aiding human annotation of sentences for training of relationship classes with large-scale language models such as BERT, which could give increased accuracy compared to GNBR scores alone.

Our overall assessment is that GNBR contains useful information that–following further curation—could be used as training sets for modern NLP models, to ultimately provide accurate, knowledge graph compatible, datasets from sentences in the scientific literature. To this end, we share our GNBR- CHEMBL merged datafile that contains over 20,000 sentences where a protein/gene-chemical co-occur and include both the GNBR relationship scores as

well as the ChEMBL (manually curated) relationships (e.g., 'agonist', 'inhibitor') —this can be accessed at https://doi.org/10.5281/zenodo.8136752. We envisage this data could be used to train a transformer model—using the CHEMBL relationship as ground truth–which would likely be very useful to aid the drug development community with future annotation efforts. Further, the GNBR relationship scores could be used as a guide to flag sentences where, for instance, the gene rather than the protein is being mentioned in a sentence. Here, the ChEMBL categories are less applicable as they refer to strictly chemical-protein interactions whereas some of the GNBR relationships such as 'increases production' could be more relevant in a chemical-gene scenario.

## Supporting information

**S1 File.**
(DOCX)

## Author Contributions

**Conceptualization:** Jonathan C. G. Jeynes.

**Formal analysis:** Jonathan C. G. Jeynes.

**Project administration:** Tim James.

**Resources:** Tim James.

**Software:** Jonathan C. G. Jeynes.

**Supervision:** Tim James.

**Visualization:** Jonathan C. G. Jeynes.

**Writing – original draft:** Jonathan C. G. Jeynes.

**Writing – review & editing:** Matthew Corney, Tim James.

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
