## [Decision Letter · Decision Letter 0]

12 May 2023

PONE-D-23-07037A large-scale evaluation of NLP-derived chemical-gene/protein relationships from the scientific literature: implications for knowledge graph constructionPLOS ONE

Dear Dr. Jeynes,

Thank you for submitting your manuscript to PLOS ONE. After careful consideration, we feel that it has merit but does not fully meet PLOS ONE’s publication criteria as it currently stands. Therefore, we invite you to submit a revised version of the manuscript that addresses the points raised during the review process.

We look forward to receiving your revised manuscript.

Kind regards,

Hilal Tayara

Academic Editor

PLOS ONE

“I have read the journal's policy and the authors of this manuscript have the following competing interests: All authors are employees of Evotec (UK) Ltd.”

Additional Editor Comments:

- make sure that the github code works well.

Reviewers' comments:

Reviewer's Responses to Questions

**Comments to the Author**

1. Is the manuscript technically sound, and do the data support the conclusions?

Reviewer #1: Yes

Reviewer #2: Yes

2. Has the statistical analysis been performed appropriately and rigorously? 

Reviewer #1: Yes

Reviewer #2: Yes

3. Have the authors made all data underlying the findings in their manuscript fully available?

Reviewer #1: Yes

Reviewer #2: No

4. Is the manuscript presented in an intelligible fashion and written in standard English?

Reviewer #1: Yes

Reviewer #2: Yes

5. Review Comments to the Author

Reviewer #1: Charlie, Jonathan, Matt, and Tim,

Congratulations on an interesting research paper. Using structured data to gain insights and potential improvements to NLP processes is an important area of research. I am recommending that PLOS publish your final version. Kindly consider my comments that follow as potential areas of improvement.

Best regards,

Richard Wendell

1. Rather than submitting this paper to “PLOS ONE,” you might consider a more targeted Journal such as PLOS Computational Biology.

2. P2 L37, Typo: the company Tellic is improperly spelled as “Telic”

3. P2 L38, in addition to AZ’s graph example, you could also choose to mention AbbVie’s Knowledge Graph that contains 911 Million Relationships (Edges). This is discussed in the paper, “Leveraging a Billion-Edge Knowledge Graph for Drug Re-purposing and Target Prioritization using Genomically-Informed Subgraphs”. doi: https://doi.org/10.1101/2022.12.20.521235

4. P3 L23-25, you might want to point out that, if I understand GNBR’s approach to assigning one relationship class per sentence, this approach can be limited when a single sentence contains multiple valid relationships.

5. P3 L37, It appears from your Github repo that the Nexis database is proprietary and may be made selectively available upon request. Your approach is reasonable here, given the importance of having a ground truth dataset for comparison. Consider making a small data slice from Nexis available as part of your submission to avoid potential issues with the publication requirements.

6. P4 L25-42, your use of fuzzy string matching is reasonable. A possible way to extend this analysis in the future would be to explore a machine-learning approach that can match chemical and gene concepts based on semantics rather than fuzzy syntactic structures.

7. P5 L29-34: Is it worth noting GNBR’s potential limitations with assigning a relationship to a binary relationship e.g. “inhibits” or “does not inhibit”….how should we treat sentences containing important auxiliary context such as “may inhibit”?

8. P6 L4, I’m not sure I agree that the reason why GNBR has fewer unique chemicals is due to “likely because NLP entity recognition for chemicals is difficult.” While I agree with the statement of the difficulty, I suspect the limited data is more likely resulting from GNBR’s approach now being five years old during a time when NLP technology has rapidly advanced. You may consider more directly referencing GNBRs limitations as a likely cause for low chemical recall.

9. P6 L8, This is an important point that you make, that GNBR’s data limitations for the chemical are partially due to the fact that most GNBR data is derived from open-access data only. Here is an interesting statistic that you could reference if you think it is relevant: “…only 28% of all scholarly publications are currently open access.” Day, S., Rennie, S., Luo, D. et al. Open to the public: paywalls and the public rationale for open access medical research publishing. Res Involv Engagem 6, 8 (2020).

10. P6 L9, Another reason why NLP-derived data can be limited is because research generally focuses on novel relationships. Thus publications only reference a few of the vast corpora of known biomedical relationships in structured sources like ChEMBL.

11. P7 L4, I don’t see the COX-1 example as a NER failure per se. NER seems to have performed correctly in assigning COX-1 a protein label. Instead, the subsequent step after NER of Entity Resolution, when an entity type (e.g. protein in this case) is then mapped to a concept ID in a known ontology (e.g. UniProt).

12. P7 L12: Similar to my previous comment, I see the Open Targets BioBERT transformer as “Entity Resolution” and not NER as stated. I would also view using a custom-trained large transformer or LLM for this NLP step as standard practice in the modern NLP approach, and thus may no longer be called “sophisticated”.

13. P10 L38: It might be worth further noting that GNBR’s use of cooccurrence is highly limited and likely generates significant false positives as many sentences have cooccurring entities that have no semantic relationship. Again, more modern NLP approaches will mitigate this challenge.

14. P11 L32: Terrific observation that GNBR could expand its data by noting binding relationships.

15. P13 L12: I agree that BERT-style language models that detect semantic structure can massively improve upon syntactic approaches like dependency parsing. You might reference tellic’s approach here as it solves this problem with such an approach: “The base model for SRQ was a large transformer-style model pre-trained on a large multi-domain corpus of scientific text and fine-tuned on expertly annotated gold standard data. The SRQ model outputs a score representing the confidence of a meaningful semantic relationship between the two entities in a sentence.” “Leveraging a Billion-Edge Knowledge Graph for Drug Re-purposing and Target Prioritization using Genomically-Informed Subgraphs”. doi: https://doi.org/10.1101/2022.12.20.521235

Reviewer #2: This paper is a study of the drug -> gene/protein edges from GNBR, a database of links between entities extracted from scientific literature. To assess the quality of these links, the authors compare them with links present in a structured database, entitled Nexus, which acts as ground truth. Essentially, one can view the contribution of this paper to be validating a subset of GNBR using a secondary resource of experimentally validated drug to gene interactions.

Knowledge graph are increasingly being used in the drug discovery domain and links derived from literature can form a great many of the links in such graphs. This paper highlights some of the potentially quality issues that can arise when using such data.

Overall I do think that this type of study could be beneficial for the community overall. I also found the paper clear to follow, well structured and I was pleased to see a section devoted to the preprocessing steps performed by the authors.

However I feel the paper still has some issues that need to be addressed before it is ready for publication. My main criticisms are:

Overall I would like to see more analysis performed in the paper and more concrete conclusions to have been drawn. In it's current state the work feels incomplete, with some good comparative work performed but more analysis needed.

There are issues with the reproducibility of this study as I was unable to run the code in the associated github repository as the "Nexus" database is not available to the public. I would prefer if the authors recreated their study with whatever parts of Nexus are open source (for example the Chembl links) to increase the reproducibility?

I feel the authors could elevate the contribution of the work by proposing a process/benchmark against which future NLP derived datasets could be compared. This work was started in the discussions section but I would like to see it expanded into a framework for future researchers to use.

The figures are low resolution meaning they appear blurry and are thus hard to read and interpret.

6. PLOS authors have the option to publish the peer review history of their article (what does this mean?). If published, this will include your full peer review and any attached files.

Reviewer #1: **Yes: **Richard Wendell

Reviewer #2: No

---

## [Author Response · Author response to Decision Letter 0]

20 Jul 2023

We would like to thank both reviewers for their very helpful and insightful comments. We have done our best to answer their queries one-by-one and feel the paper is significantly better for it. 

Reviewer #1: Charlie, Jonathan, Matt, and Tim,

Congratulations on an interesting research paper. Using structured data to gain insights and potential improvements to NLP processes is an important area of research. I am recommending that PLOS publish your final version. Kindly consider my comments that follow as potential areas of improvement.

Best regards,

Richard Wendell

1. Rather than submitting this paper to “PLOS ONE,” you might consider a more targeted Journal such as PLOS Computational Biology. 

Thanks for the suggestion. We are very happy to go with PLOS computational biology if the editors of PLOS ONE think that is favourable. 

2. P2 L37, Typo: the company Tellic is improperly spelled as “Telic”. 

We apologise for this. It is now amended. 

3. P2 L38, in addition to AZ’s graph example, you could also choose to mention AbbVie’s Knowledge Graph that contains 911 Million Relationships (Edges). This is discussed in the paper, “Leveraging a Billion-Edge Knowledge Graph for Drug Re-purposing and Target Prioritization using Genomically-Informed Subgraphs”. doi: https://doi.org/10.1101/2022.12.20.521235.

 A good example and we have now added: Another example is AbbVie/Tellic’s knowledge graph that contains 911 million relationships (edges) constructed from structured and NLP-derived data [11]. (p2L42)

4. P3 L23-25, you might want to point out that, if I understand GNBR’s approach to assigning one relationship class per sentence, this approach can be limited when a single sentence contains multiple valid relationships.

 In fact, GNBR just provides a weight for each of the ten classes that chemical-proteins sentences can be categorised as (according to their categorisation system). And you are right in that this is good when a sentence contains multiple valid relationships. It is one of the drawbacks of our approach where we are assuming that there is a “right” relationship for a chemical-protein to have, as defined by Nexus (or ChEMBL). We have tried to convey that in the discussion later on where we say: “In contrast to single label definitions, GNBR’s weighted scores across several categories could give a more nuanced representation of entity relationships.” (p12L18)

5. P3 L37, It appears from your Github repo that the Nexis database is proprietary and may be made selectively available upon request. Your approach is reasonable here, given the importance of having a ground truth dataset for comparison. Consider making a small data slice from Nexis available as part of your submission to avoid potential issues with the publication requirements.

 According to your suggestion (and that of reviewer 2) we have taken a subset of Nexus, made from only ChEMBL entries. The Github repo now runs by default using this subset of data. Plus, we have created a .csv file that contains the intersection of GNBR chemical-protein associations, with their respective equivalents from ChEMBL. Please see the response to reviewer 2 for further details. 

6. P4 L25-42, your use of fuzzy string matching is reasonable. A possible way to extend this analysis in the future would be to explore a machine-learning approach that can match chemical and gene concepts based on semantics rather than fuzzy syntactic structures. 

A good point and we have now included your comment in the manuscript on P4L43. 

7. P5 L29-34: Is it worth noting GNBR’s potential limitations with assigning a relationship to a binary relationship e.g. “inhibits” or “does not inhibit”….how should we treat sentences containing important auxiliary context such as “may inhibit”? 

This is a very important point and one we mention in the discussion (P13L20-29) where we refer to H. Kilicoglu, G. Rosemblat, and T. C. Rindflesch, “Assigning factuality values to semantic relations extracted from biomedical research literature,” PLoS One . In fact, we are experimenting with a BERT trained model from Pei et al. Measuring Sentence-Level and Aspect-Level (Un)certainty in Science Communications, https://aclanthology.org/2021.emnlp-main.784.pdf which categorises the certainty of a statement after being trained on a labelled dataset.

8. P6 L4, I’m not sure I agree that the reason why GNBR has fewer unique chemicals is due to “likely because NLP entity recognition for chemicals is difficult.” While I agree with the statement of the difficulty, I suspect the limited data is more likely resulting from GNBR’s approach now being five years old during a time when NLP technology has rapidly advanced. You may consider more directly referencing GNBRs limitations as a likely cause for low chemical recall. 

We agree and have changed our text to reflect your point. We have deleted: “likely because NLP entity recognition for chemicals is difficult.” To “There are a number of possible reasons for this. For example . . . “ 

9. P6 L8, This is an important point that you make, that GNBR’s data limitations for the chemical are partially due to the fact that most GNBR data is derived from open-access data only. Here is an interesting statistic that you could reference if you think it is relevant: “…only 28% of all scholarly publications are currently open access.” Day, S., Rennie, S., Luo, D. et al. Open to the public: paywalls and the public rationale for open access medical research publishing. Res Involv Engagem 6, 8 (2020). 

A very interesting statistic and we have included the stat and reference in our manuscript. 

10. P6 L9, Another reason why NLP-derived data can be limited is because research generally focuses on novel relationships. Thus publications only reference a few of the vast corpora of known biomedical relationships in structured sources like ChEMBL. 

A good point and one we have now included in the text.

11. P7 L4, I don’t see the COX-1 example as a NER failure per se. NER seems to have performed correctly in assigning COX-1 a protein label. Instead, the subsequent step after NER of Entity Resolution, when an entity type (e.g. protein in this case) is then mapped to a concept ID in a known ontology (e.g. UniProt). 

We agree with you here and have included your comment in the text. 

12. P7 L12: Similar to my previous comment, I see the Open Targets BioBERT transformer as “Entity Resolution” and not NER as stated. I would also view using a custom-trained large transformer or LLM for this NLP step as standard practice in the modern NLP approach, and thus may no longer be called “sophisticated”.

 You are right in that LLMs could be considered the workhorses of most NLP tasks. We have removed “sophisticated”. 

13. P10 L38: It might be worth further noting that GNBR’s use of cooccurrence is highly limited and likely generates significant false positives as many sentences have cooccurring entities that have no semantic relationship. Again, more modern NLP approaches will mitigate this challenge.

 We have added this to the text on P10:- “GNBR’s use of cooccurrence likely generates significant false positives as many sentences have cooccurring entities that have no semantic relationship. Modern NLP approaches will mitigate this challenge [11].” ( [11] is the Tellic paper. )

14. P11 L32: Terrific observation that GNBR could expand its data by noting binding relationships.

15. P13 L12: I agree that BERT-style language models that detect semantic structure can massively improve upon syntactic approaches like dependency parsing. You might reference tellic’s approach here as it solves this problem with such an approach: “The base model for SRQ was a large transformer-style model pre-trained on a large multi-domain corpus of scientific text and fine-tuned on expertly annotated gold standard data. The SRQ model outputs a score representing the confidence of a meaningful semantic relationship between the two entities in a sentence.” “Leveraging a Billion-Edge Knowledge Graph for Drug Re-purposing and Target Prioritization using Genomically-Informed Subgraphs”. doi: https://doi.org/10.1101/2022.12.20.521235. 

We have now included in the text: “Other approaches such as Tellic’s “Semantic Relationship Quantification” model – scores the confidence in a meaningful sematic relationship between two entities in a sentence – could be used to check the accuracy of relationships in individual sentences [11]. 

Reviewer #2: This paper is a study of the drug -> gene/protein edges from GNBR, a database of links between entities extracted from scientific literature. To assess the quality of these links, the authors compare them with links present in a structured database, entitled Nexus, which acts as ground truth. Essentially, one can view the contribution of this paper to be validating a subset of GNBR using a secondary resource of experimentally validated drug to gene interactions.

Knowledge graph are increasingly being used in the drug discovery domain and links derived from literature can form a great many of the links in such graphs. This paper highlights some of the potentially quality issues that can arise when using such data.

Overall I do think that this type of study could be beneficial for the community overall. I also found the paper clear to follow, well structured and I was pleased to see a section devoted to the preprocessing steps performed by the authors.

However I feel the paper still has some issues that need to be addressed before it is ready for publication. My main criticisms are:

F

Overall I would like to see more analysis performed in the paper and more concrete conclusions to have been drawn. In it's current state the work feels incomplete, with some good comparative work performed but more analysis needed. 

We have now added a concluding paragraph: “Our overall assessment is that GNBR contains useful information, that – following further curation - could be used as training sets for modern NLP models, to ultimately provide accurate, knowledge graph compatible, datasets from sentences in the scientific literature. To this end, we share our GNBR- CHEMBL merged datafile that contains over 20,000 sentences where a protein/gene-chemical co-occur, that includes both the GNBR relationship scores as well as the ChEMBL (manually curated) relationship (e.g. ‘agonist’, ‘inhibitor’) – this can be accessed https://doi.org/10.5281/zenodo.8136752. We envisage this data could be used to train a transformer model - using the ChEMBL relationship as ground truth – which would likely be very useful to aid the drug development community with future annotation efforts. Further, the GNBR relationship scores could be used as a guide to flag sentences where, for instance, the gene rather than the protein is being mentioned in a sentence. Here, the CHEMBL categories are less applicable as they refer to strictly chemical-protein interactions whereas some of the GNBR relationships such as ‘increases production’ could be more relevant in a chemical-gene scenario. 

There are issues with the reproducibility of this study as I was unable to run the code in the associated github repository as the "Nexus" database is not available to the public. I would prefer if the authors recreated their study with whatever parts of Nexus are open source (for example the Chembl links) to increase the reproducibility? 

We have now taken a subset of Nexus that only contains ChEMBL entries and have used that data as the default for the GitHub repository. You should now find that the code will run. 

I feel the authors could elevate the contribution of the work by proposing a process/benchmark against which future NLP derived datasets could be compared. This work was started in the discussions section but I would like to see it expanded into a framework for future researchers to use. 

We agree and so have provided our GNBR- ChEMBL merged datafile that contains over 20,000 sentences where a protein/gene-chemical co-occur, that includes both the GNBR relationship scores as well as the CHEMBL (manually curated) relationships (e.g. ‘agonist’, ‘inhibitor’). We have submitted this file to the data hosting site “Zenoda” (incidentally where GNBR data is held) - for easy accessibility for the community at https://doi.org/10.5281/zenodo.8136752. 

The figures are low resolution meaning they appear blurry and are thus hard to read and interpret. 

We also found that the figures in the rendered pdf are indeed very blurry, despite the fact that when the actual figure is downloaded by clicking on the hyperlink on the top right of the page, the figures are clear. Could we check with the reviewer whether they downloaded the actual figure in the manner we just explained? All our figures passed the PLOS ONE quality assessment procedure. It’s a pity that the figures don’t render clearly in the pdf despite passing the PLOS ONE assessments.

---

## [Decision Letter · Decision Letter 1]

23 Aug 2023

A large-scale evaluation of NLP-derived chemical-gene/protein relationships from the scientific literature: implications for knowledge graph construction

PONE-D-23-07037R1

Dear Dr. Jeynes,

We’re pleased to inform you that your manuscript has been judged scientifically suitable for publication and will be formally accepted for publication once it meets all outstanding technical requirements.

Kind regards,

Hilal Tayara

Academic Editor

PLOS ONE

Reviewers' comments:

Reviewer's Responses to Questions

**Comments to the Author**

1. If the authors have adequately addressed your comments raised in a previous round of review and you feel that this manuscript is now acceptable for publication, you may indicate that here to bypass the “Comments to the Author” section, enter your conflict of interest statement in the “Confidential to Editor” section, and submit your "Accept" recommendation.

Reviewer #2: All comments have been addressed

2. Is the manuscript technically sound, and do the data support the conclusions?

Reviewer #2: Partly

3. Has the statistical analysis been performed appropriately and rigorously? 

Reviewer #2: N/A

4. Have the authors made all data underlying the findings in their manuscript fully available?

Reviewer #2: Yes

5. Is the manuscript presented in an intelligible fashion and written in standard English?

Reviewer #2: Yes

6. Review Comments to the Author

Reviewer #2: (No Response)

7. PLOS authors have the option to publish the peer review history of their article (what does this mean?). If published, this will include your full peer review and any attached files.

Reviewer #2: No

---

## [Editor Report · Acceptance letter]

30 Aug 2023

PONE-D-23-07037R1 

A large-scale evaluation of NLP-derived chemical-gene/protein relationships from the scientific literature: implications for knowledge graph construction 

Dear Dr. Jeynes:

I'm pleased to inform you that your manuscript has been deemed suitable for publication in PLOS ONE. Congratulations! Your manuscript is now with our production department. 

Kind regards, 

on behalf of

Dr. Hilal Tayara 

Academic Editor

PLOS ONE